# Anatomical Study of the Ventral Upper Arm Muscles with a Case Report of the Accessory Coracobrachialis Muscle

**DOI:** 10.3390/medicina59081445

**Published:** 2023-08-10

**Authors:** Marko Vrzgula, Jozef Mihalik, Martin Vicen, Natália Hvizdošová, Ingrid Hodorová

**Affiliations:** 1Department of Anatomy, Faculty of Medicine, Pavol Jozef Šafárik University, 04180 Košice, Slovakia; 2Department of Musculoskeletal and Sports Medicine, AGEL Hospital Košice-Šaca, 04015 Košice-Šaca, Slovakia

**Keywords:** anatomic variation, arm, dissection, muscles

## Abstract

*Background and Objectives:* The muscles in the upper arm are categorized into two groups: ventral muscles, which include the biceps brachii, coracobrachialis, and brachialis, and dorsal muscles comprising the triceps brachii and anconeus. These muscles are positioned in a way that they contribute to movements at the shoulder and elbow joints. Given the importance of the upper arm muscles for various reasons, they need to be well-known by medical professionals. Ventral upper arm muscles exhibit various topographical and morphological variations. Understanding these variations is critical from both anatomical and clinical standpoints. Therefore, our aim was to conduct an anatomical study focusing on these muscles and potentially identify ventral upper arm muscle variations that could contribute to the broader understanding of this area. For this anatomical study, 32 upper limbs obtained from 16 adult cadavers were dissected. *Case report:* During our anatomical survey, an accessory coracobrachialis muscle in the left upper extremity of one cadaver was discovered. This additional muscle was located anterior to the classical coracobrachialis muscle and measured 162 mm in length. It originated from the distal anterior surface of the coracoid process and was inserted into the middle third of the humeral shaft. The accessory muscle was supplied by the musculocutaneous nerve. No apparent anatomic variations were observed in the other upper arm muscles in any of the cadavers. *Conclusions:* Gaining insight into the ventral upper arm muscle variations holds vital significance in both anatomy and clinical practice, as they can influence surgical approaches, rehabilitation strategies, and the interpretation of imaging studies. Based on the morphological characteristics of the accessory coracobrachialis muscle discovered in our case, we hypothesize that it could have caused an atypical palpable mass in the medial brachial area, adjacent to the short head of the biceps brachii.

## 1. Introduction

The upper arm muscles are a group of muscles in the upper extremity. They are widely recognized among clinicians, particularly orthopedists, traumatologists, rehabilitation physicians, as well as specialists who perform anesthesia techniques. The upper arm muscles, at least some of them, are also known even among the general population because they are frequently used in everyday activities. Additionally, these muscles are easily accessible for dissection. Consequently, we have decided to conduct a detailed anatomical study specifically focusing on the ventral upper arm muscles to explore any anatomic variations and contribute to the expansion of knowledge in this area.

Undertaking this research enables a comprehensive understanding of these muscles and their potential variations, which, in turn, proves essential in various fields, including anatomy, upper limb surgery, and radiology, as it empowers clinicians to tailor diagnostics and treatments more effectively.

The muscles of the upper arm work together to allow for various motions, including flexion, extension, abduction, adduction, and rotation of the arm. They surround the humerus and have a partial origin from it, although some also originate from the shoulder girdle. Their primary insertion is into the bones of the forearm, spanning the elbow joint. The upper arm muscles are responsible for flexion or extension in this joint, dividing them into two main functional groups. The flexor group is located on the ventral side and includes the biceps brachii (BB) muscle. The BB has two heads that also play a secondary role in the shoulder joint. The second ventral muscle of the upper arm is the brachialis, which lies beneath the BB. The third ventral upper arm muscle is the coracobrachialis (CB), which is considered part of the flexor group due to its location and innervation. Unlike other muscles in the region, the CB does not directly affect the movement of the elbow joint, but provides assistance to the motion of the shoulder joint [1].

The dorsal aspect of the upper arm is composed of the triceps brachii (TB) and the anconeus muscles. The TB consists of three heads. The long head starts at the scapula, specifically at the infraglenoid tubercle, while the lateral and medial heads originate from the dorsal surface of the humerus. These three heads converge and insert together at the olecranon. The primary function of TB is to serve as the major extensor of the elbow joint. While the medial and lateral heads primarily impact the movement of the elbow, the long head additionally contributes to movements in the shoulder joint, such as retroversion and adduction [2]. The anconeus muscle extends from the lateral epicondyle to both the olecranon and the posterior side of the ulna. However, it has limited functional significance compared to other muscles in the area [3].

All ventral muscles of the upper arm are innervated by the musculocutaneous nerve; it emerges from the lateral cord of the brachial plexus. The median nerve is created from the ventral branches of cervical spinal nerves, ranging from the fifth to the seventh. On the other hand, the dorsal muscles of the brachium are supplied by the radial nerve. It originates from the posterior cord of the brachial plexus and includes fibers from the nerves derived from the fifth to the eighth cervical spinal cord segments, as well as the first thoracic spinal nerve [4].

The ventral group of upper arm muscles is separated from the dorsal group by the lateral and medial intermuscular septum of the arm. The origin of the lateral intermuscular septum lies at the crest of the greater tubercle and extends toward the lateral epicondyle [5]. In contrast, the medial intermuscular septum is thicker compared to the lateral one and is located between the crest of the lesser tubercle, just below the insertion tendon of the teres major muscle and the medial epicondyle [6].

## 2. Detailed Case Description

The anatomical study was realized at the Faculty of Medicine of Pavol Jozef Šafárik University in Košice. It was conducted on thirty-two upper limbs from sixteen adult cadavers (47–89 years), which were used during undergraduate dissections for first-year general medicine students. All cadavers were obtained through the body donation program, with proper informed consent from the donors, and were under the administration of the Department of Anatomy. Since the cadavers were solely used for educational and research purposes, ethical clearance was not required. Among the cadavers, there were eight males and eight females. The cadavers showed no evident pathological conditions and had been preserved with a formalin-based solution shortly after death.

For our piece of research, the skin with the underlying subcutaneous soft tissue at the upper arm level was removed. Superficial neurovascular structures and the brachial fascia were revealed, and further dissection was realized to identify all ventral muscles of the upper arm, along with adjacent nerves and vessels. During the dissection, the origins and insertions of the ventral upper arm muscles were observed, and the course of the neurovascular structures was recorded. Metric analyses were conducted using a sliding caliper, with an estimated measurement error of 2 mm.

### Case Report

Sixteen right and sixteen left upper arms from eight male and eight female cadavers were examined to investigate the ventral upper arm muscles. The origins and insertions of all ventral upper arm muscles were identified, and their topographical and morphological characteristics were noted in all dissected upper extremities with the identification of one anatomic variation.

During the examination, a noteworthy finding of an accessory CB muscle was discovered in the left upper extremity of a 74-year-old female cadaver (Figure 1). This muscle had only a short proximal tendon and started from the distal anterior surface of the coracoid process (CP), medially to the origin of the short head of the BB (Figure 2). It had a distinct and well-developed muscular body, contrasting with the thin and longer BB short head proximal tendon, which was positioned laterally and was isolated (Figure 3). Behind the short head of the BB and accessory CB, another isolated muscle was situated, which also had its origin at the CP. This muscle corresponds to the conventional description of the CB (Figure 4 and Figure 5). It measured 143 mm in length and was inserted into the central third of the medial margin of the humeral shaft. In comparison, the accessory CB measured 162 mm in length and was attached to a more distal location on the humeral shaft, to be more exact, medially to the origin of the brachialis muscle (Figure 6). Both CB muscles were innervated by thin branches originating from the musculocutaneous nerve, with the nerve penetrating the conventional one. Important to note is that these muscles did not compress the nearby neurovascular structures within the upper arm region and their normal positioning remained unaltered.

Furthermore, no additional variations were found in the upper arm muscles, neither in this cadaver nor any of the other cadavers.

## 3. Discussion

To ensure the accurate performance of diverse medical procedures, healthcare professionals must be familiar not only with standard muscle anatomy, but also with potential muscle variations, as these can significantly impact the appropriate approach to both diagnostic and therapeutic interventions. Additionally, muscle variations can give rise to various clinical disorders.

In general, variations in muscle anatomy encompass a wide range of anatomical differences that can be observed among individuals. Anatomic variations in muscles, such as their size, shape, and arrangement, can impact the surrounding nerves and blood vessels [7]. For instance, compressed or entrapped nerves due to anatomical irregularities may contribute to neuropathy. Symptoms of neuropathy vary and include alterations in feeling, muscle wasting, discomfort, loss of sensation, and even disruptions in autonomic functions [8]. Similarly, abnormal blood vessel pathways or reduced blood flow caused by anatomical factors could be linked to vasculopathy. Given the critical role of blood vessels in transporting nutrients, waste products, and oxygen throughout the body, vascular dysfunction has the potential to lead to ischemia [9].

Identifying and characterizing these anatomical connections is crucial for diagnosing and managing neuropathy or vasculopathy effectively. Moreover, this awareness can aid in developing targeted therapy strategies, including surgical interventions, to release nerve compression or address vascular abnormalities.

The ventral upper arm muscles form a muscle group that plays a vital role in the movement and stability of the arm, particularly at the shoulder and elbow joints. The contour of the anterior surface of the brachium is determined by the BB. It is a superficially positioned, double-headed, and double-jointed muscle. Both heads originate near the shoulder joint at the scapula and merge into a single belly. The long head starts at the supraglenoid tubercle just above the glenoid cavity. The long head tendon passes through the shoulder joint cavity and is covered by a synovial sheath. Distally, the tendon runs into the intertubercular sulcus, where the synovial sheath extends caudally as the intertubercular tendon sheath. The distal part of the long head tendon then continues from the lower part of the intertubercular groove into the belly of the BB, no longer being covered by a synovial membrane [10]. The short head is located medially to the long head. It originates at the CP in front of the CB muscle. The primary insertion of the BB is at the radial tuberosity. Between the tendon and the radial tuberosity is positioned the bicipitoradial bursa. Alongside that, the BB inserts into the bicipital aponeurosis, which is fused with the forearm fascia. The principal functions of the BB are flexion and supination at the elbow joint. It is the most important supinator of the flexed elbow joint and the strongest flexor at the elbow. The secondary function of the long head is abduction and medial rotation in the shoulder joint, while the secondary function of the short head is the adduction of the shoulder joint [11].

The BB exhibits diverse anatomic variations that are capable of affecting any segment of the muscle. The origins of the long head and short head can be localized differently on the scapula or around the shoulder joint. The extent to which the muscle is split into its heads also varies individually. In extreme cases, complete division of the muscle may occur. An accessory head of the BB can originate from the scapula, humerus, medial and lateral intermuscular septum, surrounding muscles, and soft tissues around the shoulder joint or the joint capsule itself. The third head has the ability to join the long head, short head, common belly of the muscle, or the bicipital aponeurosis variably. Finding a greater number of accessory heads is rare. The mentioned BB variations have been extensively documented in numerous studies [7,12,13,14,15].

The brachialis originates from the distal ventral aspect of the humerus and extends toward the ulnar tuberosity. It is situated beneath the BB and exemplifies a typical flexor muscle [16]. If the brachialis and BB fail, often due to damage to the musculocutaneous nerve, elbow flexion is greatly affected. However, there is still some limited bending ability due to the remaining forearm flexor muscles (supplied by the median nerve) and the lateral forearm muscles (supplied by the radial nerve) [11].

Traditionally, brachialis has been characterized as a muscle with a single head. Nevertheless, investigations conducted on cadavers have revealed that the brachialis muscle may, in fact, consist of two heads, namely a superficial head and a deep head [17].The anatomic variations of the brachialis can manifest as accessory slips that originate from nearby structures and merge with the main muscle. Less frequent variations occur at the insertion site, where the muscle may, for instance, connect to the bicipital aponeurosis or radius [18].

CB is located underneath the BB, more precisely, its short head. It starts on the CP and inserts into the inner surface of the middle third of the humeral shaft. Its primary functions are anteversion, internal rotation, and adduction in the shoulder joint [1]. The CB serves as an important anatomical landmark for the brachial plexus, as the musculocutaneous nerve typically passes through its muscle belly [19].

Anatomic variations in the CB are commonly associated with the presence of supernumerary heads, which can be categorized into two groups for easier classification. The insertion point of the CB brevis is located in the proximal part of the humerus, while the CB longus is attached to the lower portion of the upper arm [20]. Another type of CB variation involves complete or incomplete separation of the muscle belly. In Mori’s study on the Japanese population, the separation of the CB muscle into its two layers, namely the superficial and deep layers, was incomplete in 8% of cases and complete in 16% of cases [21]. In this study, unfortunately, there was no image documentation, and a more detailed description of the cases was also missing. As a result, the accessory muscle observed in our case cannot be classified as either the CB brevis or longus. Instead, we consider it to be a duplication or complete separation of the muscle.

Understanding the variations in muscles can be gained by studying the embryological formation of the muscle. The upper limb muscles differentiate locally from the mesenchyme of the limb bud, which originates from the mesoderm of the lateral plate. During a specific stage of development, the muscle primordia present in distinct layers of the upper limb merge to form a cohesive muscle mass. Subsequently, some muscle primordia naturally undergo cell death and disappear. The existence of an accessory CB could be attributed to the early termination of the natural regression process [22].

The CB muscle actively participates in various upper extremity movements and helps maintain stability and control in the shoulder region. However, its location and proximity to the surrounding neurovascular structures can complicate surgical access, potentially requiring modifications to the surgical approach. Thus, variations in muscle anatomy hold clinical importance for surgeons and also radiologists who analyze both traditional and computerized imaging techniques [23].

In terms of surgical applications, the CB has the potential to fulfill various functions. It can be utilized as a transposition flap in the correction of deformities in the subclavicular and axillary regions during post-mastectomy reconstruction. In addition, it can be employed as a vascularized muscle transfer in treating chronic facial paralysis. Moreover, the CB serves as a helpful reference for identifying the axillary artery during upper arm surgery and anesthesia administration [19].

## 4. Conclusions

From a clinical perspective, variations in muscles can cause a palpable mass or disrupt the normal course of neurovascular structures. Moreover, surgical planning can be challenging when muscle variations are present, as they have the potential to hinder the identification of neurovascular structures within the muscular compartments.

In specific cases, muscle variations can result in neurovascular compression, leading to subsequent neuropathy or vasculopathy.

Considering that our study was retrospective, we did not have access to the donors‘ clinical data regarding potential neuropathy or vasculopathy in the left upper limb. However, we assumed that such issues were not present in the donor since we did not find any anomalies of nerves and blood vessels in that region during the autopsy.

On the other hand, we presume that the accessory CB muscle could have caused an atypical palpable swelling in the left upper arm, medial to the short head of the BB. In a clinical examination, such a palpable mass can potentially be misinterpreted as lymphadenopathy. Therefore, the presence of an accessory CB, as discovered in our case, may complicate the surgical and anesthesia techniques in the upper arm.

Overall, understanding the function and anatomic variations in the CB muscle is essential for clinicians in planning and executing procedures in the brachial region.

## Figures and Tables

**Figure 1 medicina-59-01445-f001:**
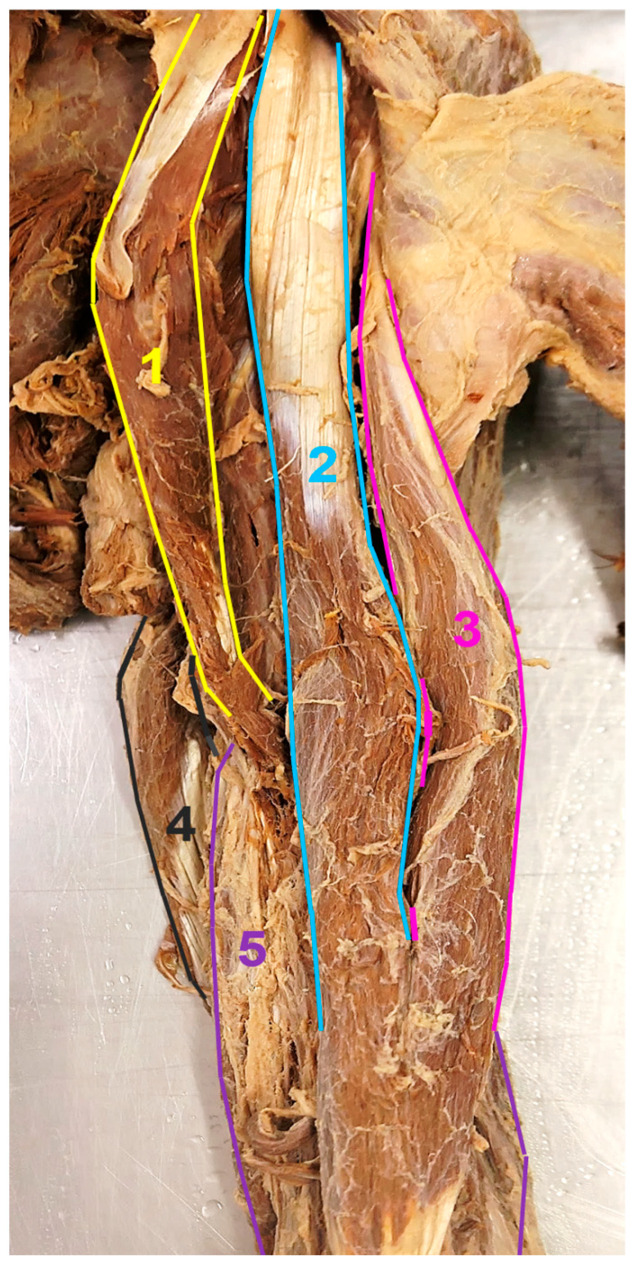
Left upper arm muscles. Anterior view: 1—accessory CB, 2—short head of the BB, 3—long head of the BB, 4—medial head of the triceps brachii, 5—brachialis.

**Figure 2 medicina-59-01445-f002:**
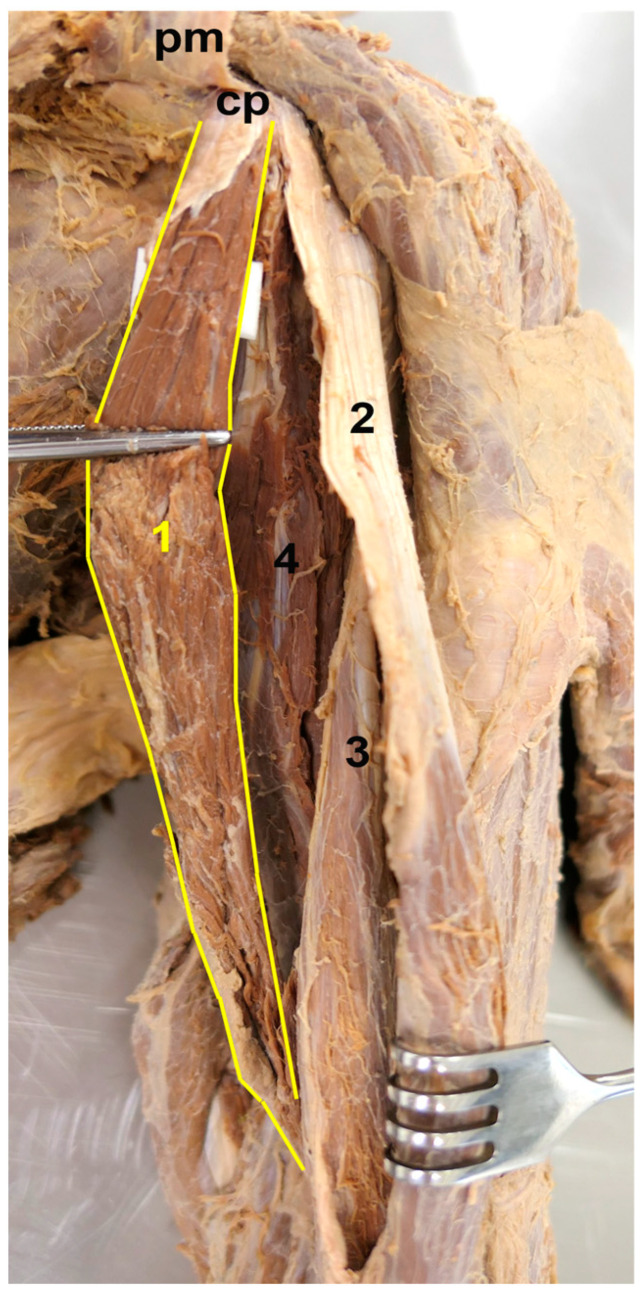
The origin of the accessory CB. Anteromedial view. In order to achieve a clearer view, the accessory CB was pulled slightly medially, the short head of the BB was pulled slightly laterally, and the pectoralis minor (pm) was pulled superiorly. 1—accessory CB, 2—short head of the BB, 3—long head of the BB, 4—conventional CB, cp—coracoid process.

**Figure 3 medicina-59-01445-f003:**
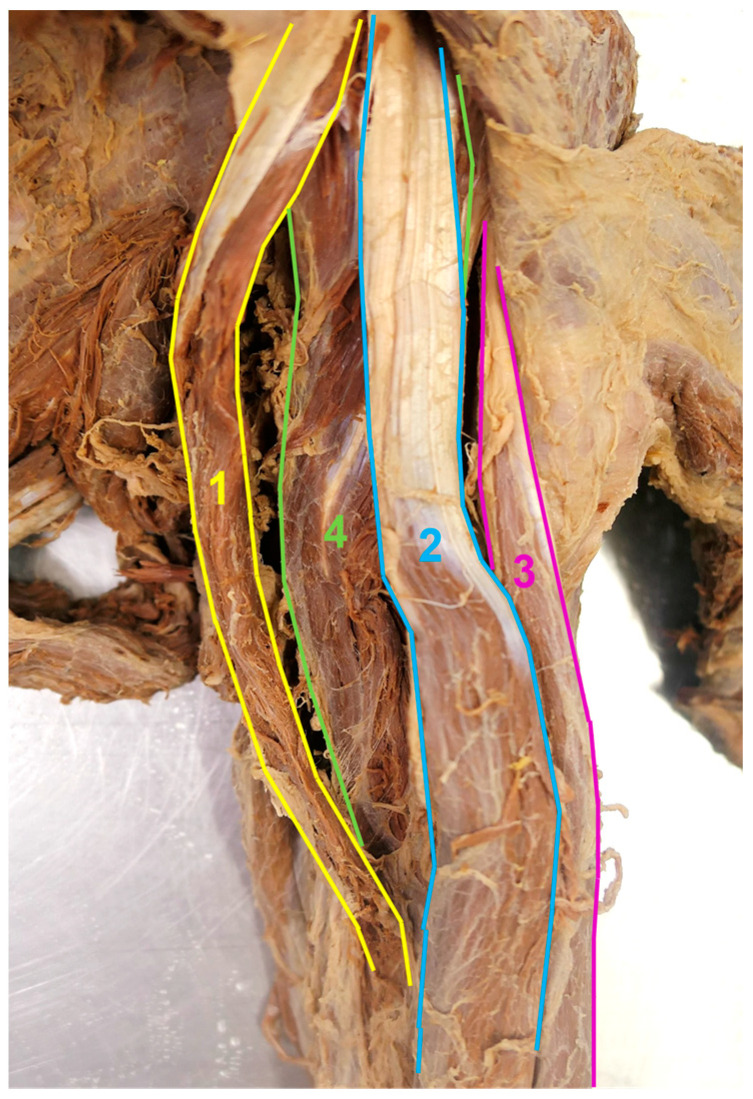
Left ventral upper arm muscles. Anteromedial view. The accessory CB was pulled medially for a better identification of the short head of the BB and the conventional CB. 1—accessory CB, 2—short head of the BB, 3—long head of the BB, 4—conventional CB.

**Figure 4 medicina-59-01445-f004:**
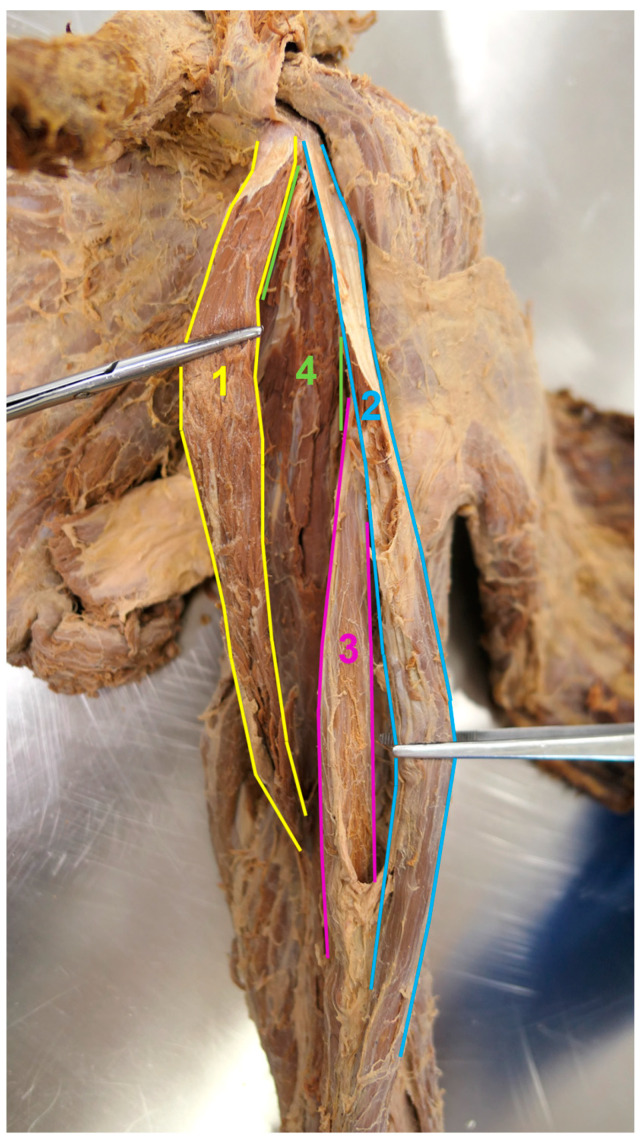
The anteromedial view behind the accessory CB and the isolated thin proximal tendon of the short head of the BB. The accessory CB was pulled medially and the short head of the BB was pulled laterally with the aim of better visualizing the conventional CB. 1—accessory CB, 2—short head of the BB, 3—long head of the BB, 4—conventional CB.

**Figure 5 medicina-59-01445-f005:**
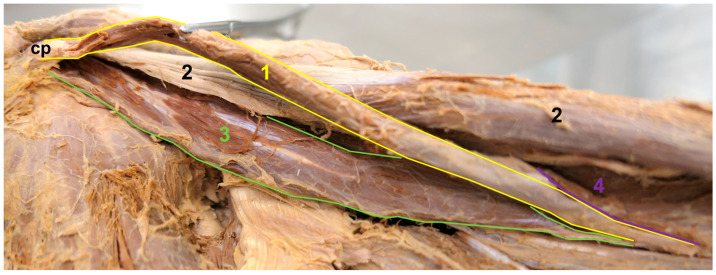
The medial view behind the accessory CB. The accessory CB was pulled slightly anteriorly for a better visualization of the conventional CB. 1—accessory CB, 2—short head of the BB, 3—conventional CB, 4—brachialis, cp—coracoid process.

**Figure 6 medicina-59-01445-f006:**
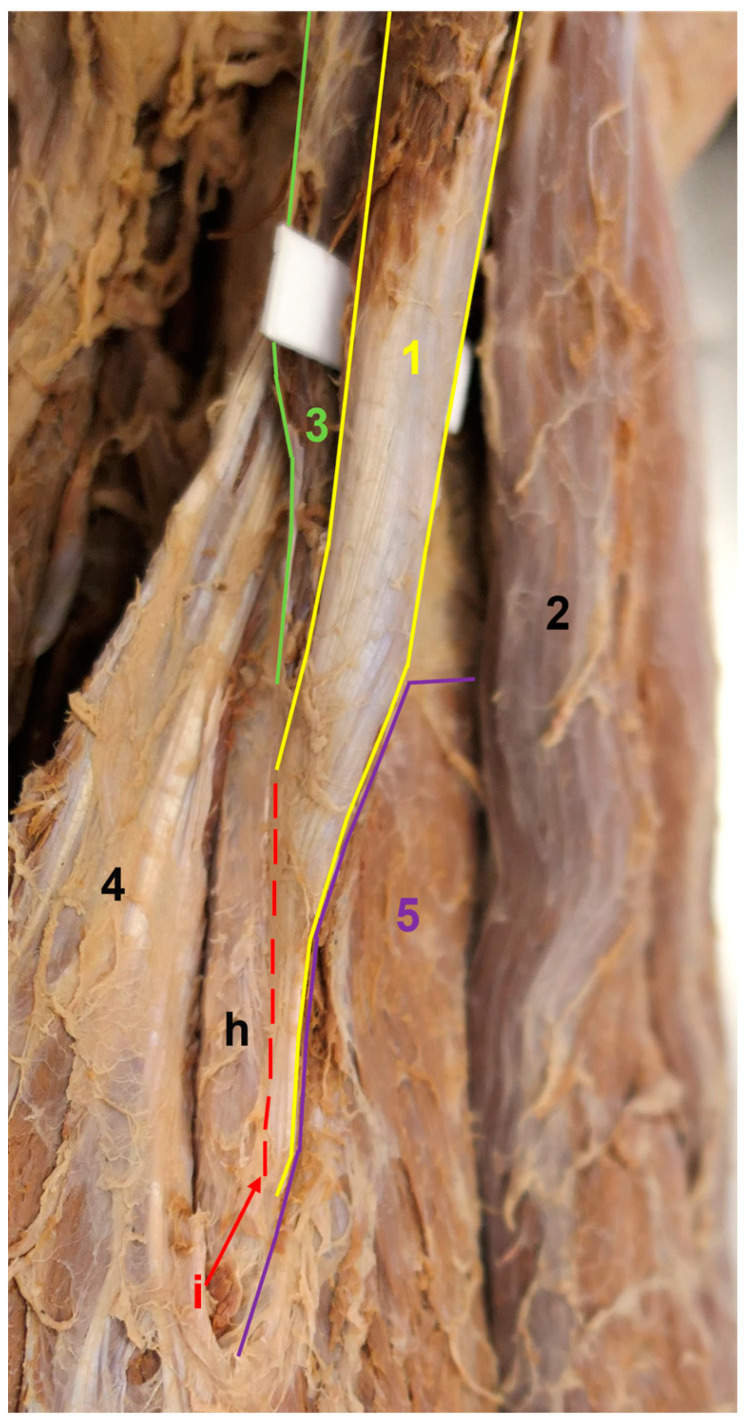
The insertion of the accessory CB. Medial view. The medial head of the triceps brachii was pulled medially and dorsally to enhance the visualization of the accessory CB insertion. 1—accessory CB, 2—short head of the BB, 3—conventional CB, 4—medial head of the triceps brachii, 5—brachialis, i—insertion of the accessory CB, h—humerus.

## Data Availability

Data are contained within the article.

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
