# Peer review of "Anatomical Study of the Ventral Upper Arm Muscles with a Case Report of the Accessory Coracobrachialis Muscle"

_medicina, 2023, doi:10.3390/medicina59081445_

Round 1
Reviewer 1 Report
The case report is interesting, especially considering that the authors should review crucial and relevant points for publication. Among these, they should provide more illustrations, photographs, of the dissection, focusing on the specific muscle. This should include its origin and insertion, adequately isolating and improving the dissection of its belly from the surrounding tissues. Furthermore, the authors should revise the presentation of the brachial biceps muscle and its heads in Figures 1 and 2. As highlighted throughout the text, the authors emphasize the possibility of neurovascular compression. However, this justification does not align with or add anything to the report. Therefore, the authors should substantially revise the information in the case report to enhance its quality and scientific contribution.

The case report is interesting, especially considering that the authors should review crucial and relevant points for publication. Among these, they should provide more illustrations, photographs, of the dissection, focusing on the specific muscle. This should include its origin and insertion, adequately isolating and improving the dissection of its belly from the surrounding tissues. Furthermore, the authors should revise the presentation of the brachial biceps muscle and its heads in Figures 1 and 2. As highlighted throughout the text, the authors emphasize the possibility of neurovascular compression. However, this justification does not align with or add anything to the report. Therefore, the authors should substantially revise the information in the case report to enhance its quality and scientific contribution.
Author Response
Response to Reviewer 1 Comments
Point 1: Authors should provide more illustrations, photographs, of the dissection, focusing on the specific muscle. This should include its origin and insertion, adequately isolating and improving the dissection of its belly from the surrounding tissues.
Response 1: We revised and adjusted all figures based on your recommendations. For better orientation, we added illustrated borders outlining the muscles in all figures and included 5 additional photographs taken after improved dissection to facilitate better positioning. We also incorporated images of the accessory muscle's origin and insertion from different angles. To enhance readability, we added viewing directions to all figures. Moreover, we revised the figure descriptions to provide more detail.
During the dissection, we endeavoured to remove all subcutaneous tissue. On the surface of the muscles, small remnants of the fascial sheath remained, which were challenging to remove without damaging the muscle fibers, unfortunately, in a few places, we were not successful.
We firmly hope that this will not significantly compromise the quality of the images and that individual muscles will be well distinguishable from each other in the revised version of the manuscript.
Point 2: Furthermore, the authors should revise the presentation of the brachial biceps muscle and its heads in Figures 1 and 2.
Response 2: We revised the presentation of the biceps brachii muscle and its heads in the previous Figure 1. We removed the previous Figure 2. Additionally, we have added new figures illustrating the biceps brachii from different views to enhance spatial awareness.
Point 3: As highlighted throughout the text, the authors emphasize the possibility of neurovascular compression. However, this justification does not align with or add anything to the report.
Response 3:
Since several findings of neurovascular compression due to variations in muscle anatomy have been described in the literature, we investigated the potential presence of neurovascular compression in our case as well.
Considering that our study was retrospective, we lacked access to the donor's clinical data concerning possible vasculopathy or neuropathy in the left upper limb region. Nevertheless, we assumed that such issues were absent in the donor as we did not observe any abnormalities of blood vessels and nerves in that area during the autopsy.
Consequently, we provided a more detailed explanation and reflection on this observation in the conclusions section of the manuscript, which was revised.
Furthermore, in the discussion section of the manuscript, we included general information about the etiopathogenesis of neurovascular compression in correlation with anatomical variations of muscles. This consideration is essential in clinical practice, correlates with the aim of the Special Issue, and it was also added based on the suggestion of another reviewer. In comparison with the previous version of the manuscript, we additionally cited Hammi and Yeung (2023) and Tucker et al. (2023) to supplement this information.
In contrast, we have removed the details about neurovascular compression from the abstract of the manuscript, which was also revised.
Thank you for your valuable feedback on our anatomical study. We truly appreciate both the overall review and all the comments on the manuscript. We have carefully considered and incorporated all your suggestions during the editing process. Based on your recommendations, we have made revisions throughout the entire manuscript.

Reviewer 2 Report
The article is well written. However, the research question that lead the authors to conduct this study could be explained in more detail. E. g., "Upper arm muscle variations encompass a wide range of anatomical differences that can be observed among individuals." (line 68). The majority of the introduction is dedicated to the explanation of the anatomy of the arm; if the authors explained in more detail the correlation between the anatomy and the etiopathogenesis of neuropathy or vasculopathy the aim of the study would be more clearly explained
Author Response
Response to Reviewer 2 Comments
Point 1: The research question that lead the authors to conduct this study could be explained in more detail.
Response 1: The authors' reason for conducting this study is explained in more detail below and has been included in the revised version of the manuscript :
The upper arm muscles are widely recognized not only among medical professionals but usually also by the general population. They are frequently used in everyday activities and are easily accessible for dissection. Consequently, our primary objective was to conduct a detailed anatomical study specifically focusing on these muscles to explore any anatomical variations and contribute to the expansion of knowledge in this area. Undertaking this research enables a comprehensive understanding of upper arm muscles and their potential variations, which, in turn, proves essential in various fields, including anatomy, orthopedics, traumatology, and radiology, as it empowers clinicians to tailor diagnostics and treatments more effectively
Point 2: The majority of the introduction is dedicated to the explanation of the anatomy of the arm; if the authors explained in more detail the correlation between the anatomy and the etiopathogenesis of neuropathy or vasculopathy the aim of the study would be more clearly explained.
Response 2: The correlation between the anatomy and the etiopathogenesis of neuropathy or vasculopathy is explained in more detail below and has been included in the revised version of the manuscript.
In comparison with the previous version of the manuscript, we additionally cited Hammi and Yeung (2023) and Tucker et al. (2023) to supplement this information.
Anatomic variations in muscles, such as their size, shape, and arrangement, can impact the surrounding nerves and blood vessels. For instance, compressed or entrapped nerves due to anatomical irregularities may contribute to neuropathy. Symptoms of neuropathy vary and include alterations in feeling, muscle wasting, discomfort, loss of sensation, and even disruptions in autonomic functions. Similarly, abnormal blood vessel pathways or reduced blood flow caused by anatomical factors could be linked to vasculopathy. Given the critical role of blood vessels in transporting nutrients, waste products, and oxygen throughout the body, vascular dysfunction has the potential to lead to ischemia.
Identifying and characterizing these anatomical connections is crucial for diagnosing and managing neuropathy or vasculopathy effectively. Moreover, this awareness can aid in developing targeted treatment strategies, including surgical interventions to release nerve compression or address vascular abnormalities.
Thank you for your valuable feedback on our anatomical study. We truly appreciate both the overall review and the comments on the manuscript. We have carefully considered and incorporated your suggestions to the manuscript during the editing process.

Round 2
Reviewer 1 Report
Accept